# Evaluation of Safety and Immunogenicity of a Recombinant Receptor-Binding Domain (RBD)-Tetanus Toxoid (TT) Conjugated SARS-CoV-2 Vaccine (PastoCovac) in Recipients of Autologous Hematopoietic Stem Cell Transplantation Compared to the Healthy Controls; A Prospective, Open-Label Clinical Trial

**DOI:** 10.3390/vaccines11010117

**Published:** 2023-01-03

**Authors:** Maryam Barkhordar, Mohammad Ahmadvand, Leyla Sharifi Aliabadi, Seied Saeid Noorani, Fahimeh Bagheri Amiri, Ghasem Janbabai, Rahim Sorouri, Mona Asadi Milani, Mohammad Vaezi

**Affiliations:** 1Cell Therapy and Hematopotic Stem Cell Transplantation Research Center, Research Institute for Oncology, Hematology and Cell Therapy, Tehran University of Medical Sciences, Tehran 14155-6559, Iran; 2Department of Epidemiology and Biostatistics, Research Centre for Emerging and Reemerging Infectious Diseases, Pasteur Institute of Iran Tehran 1316943551, Iran; 3Pasteur Institute of Iran, Tehran 1316943551, Iran

**Keywords:** hematopoietic stem cell transplantation, RBD-TT conjugated, SARS-CoV-2 vaccine, immune response

## Abstract

**Background**: The urgent need for prompt SARS-CoV-2 immunization of hematopoietic stem cell transplant (HSCT) recipients in an endemic area raises many challenges regarding selecting a vaccine platform appropriate for HSCT recipients being economical for widespread use in developing countries. **Methods**: The trial is a prospective, single-group, open-label study to investigate the safety and serologic response of two doses of the recombinant receptor-binding domain (RBD)-Tetanus Toxoid (TT) conjugated SARS-CoV-2 vaccine (PastoCovac) early after autologous (auto) HSCT. For this reason, a total of 38 patients who completed the two-dose SARS-CoV-2 RBD-based vaccine between three to nine months after auto-HSCT and had an available anti-spike serologic test at three predefined time points of baseline and after the first and second doses and 50 healthy control individuals were included in the analysis. The primary outcome was defined as an increase in IgG Immune status ratio (ISR) to the cut-off value for the positive result (≥1.1) in the semiquantitative test. **Findings**: The median time between auto-HSCT and vaccination was 127 days. No participant reported any significant adverse effects (Grade 3). Pain at the injection site was the most common adverse event. The ISR increased significantly (*p* < 0.001) during the three-time point sampling for both patients and healthy control groups. In patients, the mean ISR increased from 1.39 (95% CI: 1.13–1.65) at baseline to 2.48 (1.93–3.03) and 3.73 (3.13–4.38) following the first and second dosages, respectively. In multivariate analysis, the higher count of lymphocytes [OR: 8.57 (95% CI: 1.51–48.75); *p* = 0.02] and history of obtaining COVID-19 infection before transplantation [OR: 6.24 (95% CI: 1.17–33.15); *p* = 0.03] remained the predictors of the stronger immune response following two doses of the RBD-TT conjugated vaccine. Moreover, we found that the immunogenicity of the COVID-19 vaccine shortly after transplantation could be influenced by pre-transplant COVID-19 vaccination. **Interpretation**: The RBD-TT conjugated SARS-CoV-2 vaccine was safe, highly immunogenic, and affordable early after autologous transplants. **Funding**: This work was mainly financed by the Hematology-Oncology-Stem Cell Transplantation Research Center (HORCSCT) of Tehran University and the Pasteur Institute of Iran.

## 1. Background

A big medical emergency has been declared due to the new coronavirus disease 2019 (COVID-19) caused by the severe acute respiratory syndrome virus (SARS-CoV-2). Several studies have found that in the context of the COVID-19 pandemic, hematopoietic stem cell transplant (HSCT) recipients are more likely than the general population to develop the most severe symptoms of SARS-CoV-2 infection and have a higher fatality rate [1,2]. Despite the concept that immune responses to vaccination are frequently limited and unreliable early after HSCT, many scientific societies, notably the European Society for Blood and Marrow Transplantation (EBMT), recommend immunization as early as three months following HSCT to elicit early protective immunity in this high-risk patient population [3,4].

It is also noteworthy that the serological response to the two doses of the SARS-CoV-2 vaccine was significantly reduced within the first few months following HSCT [5,6,7,8]; however, this varies on the kind of HSCT, underline malignancy, or vaccine utilized. There is no preferred COVID-19 vaccine for transplanted patients. However, mRNA vaccines (BNT162b2 from Pfizer-BioNTech and mRNA-1273 from Moderna) and adenoviral vector vaccines (Ad26.COV2.S from Johnson & Johnson and ChAdOx1 from AstraZeneca) were the most often used vaccine platform in HSCT recipients [5,6].

The alternative vaccine technology, which is based on SARS-CoV-2 protein components such as spike protein (S1) and receptor-binding domain (RBD), is the protein subunit platform, one that has exhibited benefits in terms of safety, immunogenicity, and affordability [9]. Available studies indicate that RBD-based SARS-CoV-2 vaccines, such as NovoVax, Zhifei, and Noora vaccines, provide encouraging results in normal subjects [10,11,12,13]. By coupling RBD to tetanus toxoid (TT), humoral and cellular immune responses were enhanced more robustly [14]. Soberana 2 or PastoCovac is a recombinant RBD conjugated to TT that was developed jointly by the Cuban Finlay Institute (known as Soberana 2) and the Iranian Pasteur Institute (named PastoCovac) [15]. The vaccine has been approved for emergency use in Cuba, where it is also licensed for children older than two years. In addition, emergency use permission for the vaccine has been granted in Iran.

The need for access to an effective and affordable vaccine for HSCT recipients in the endemic area, we conducted this prospective study to evaluate the safety and immunogenicity of two doses of a different vaccine platform (RBD-TT conjugated SARS-CoV-2 vaccine) early after autologous (auto-) HSCT compared to the healthy controls. We also examined whether recipients’ immune responses to early post-HSCT immunization were affected by pre-HSCT vaccination, the interval between HSCT and vaccination, and the basic characteristic of patients.

## 2. Methods

### 2.1. Study Design and Participants

Our study is a single-group, open-label clinical trial to evaluate the safety and efficacy of two doses of the RBD-TT conjugated SARS-CoV-2 vaccine (PastoCovac) in the early post-transplant period in adult patients undergoing autologous HSCT at the Hematology-Oncology-Stem Cell Transplantation Research Center (HORCSCT) of Tehran University in collaboration with the Pasteur Institute of Iran. The study was registered with the Iranian Registry of Clinical Trials (IRCT20140818018842N23) and ClinicalTrial.gov (NCT05185817). Recruitment for the trial began in April 2022.

### 2.2. Inclusion Criteria

Recipients of auto-HSCT, more than 18 years of age, who were transplanted during the last three to twelve months, with successful neutrophil and platelet engraftment, and no known history of SARS-CoV-2 infection following HSCT, were all included in the study. The control group comprised fifty healthy adults without substantial co-morbidities or immunocompromised conditions.

### 2.3. Exclusion Criteria

The exclusion criteria included: recognized SARS-CoV2 infection within the previous three months, recurring post-HSCT infection, receiving rituximab within the previous six months, coagulation disorder or severe thrombocytopenia which contravened an intramuscular injection, history of an allergic response to the active compounds in the vaccine, inability to provide consent forms, ongoing graft rejection or relapse of underlying disease.

### 2.4. Regulatory and Ethical Approval and Written Informed Consent

The trial was performed under the Declaration of Helsinki and Good Clinical Practice. It was approved by the Ethics Committee of the Hematology-Oncology and Stem Cell Transplantation Research Center at Tehran University (IR.TUMS.HORCSCT.REC.1400.035). Each subject provides informed consent to administer the PastoCovac vaccine and collect blood samples.

### 2.5. Procedures

The flow chart of study selection was presented in detail in Figure 1. Beginning in April 2022, seventy auto-HSCT recipients were recruited. Forty-nine participants met the inclusion criteria and remained in the experiment to get post-HSCT immunization. The primary causes for screening failure were post-HSCT COVID-19 infection, refusal to participate, and the use of a different type of vaccination. Consequently, 38 patients who obtained the two-dose SARS-CoV-2 RBD-based vaccine between three and nine months after auto-HSCT had an anti-spike serologic test at three-time points of baseline and after the first and second doses were included in the final analysis.

Before the first injection and three weeks (±7 days) following each dose of vaccine, 2 mL of the patient’s venous peripheral blood was taken and delivered to the laboratory to evaluate the immune response by a semiquantitative anti-spike serologic test. The trained medical staff administered vaccination which consisted of 0.5 mL of vaccine given intramuscularly in the deltoid region, per manufacturer guidelines. Using the web-based software of our institution, the researcher creates an electronic case report form (CRF) to record study data, including baseline characteristics, concomitant drugs, vaccine-related events, lymphocyte count, and SARS-CoV-2 anti-S1 titers.

As a control group, we enrolled 50 healthy volunteers (22 female, 28 male), with a mean age of 37.92 (SD = 12.62) and no evidence of SARS-CoV-2 infection before to immunization, who were vaccinated with two doses of RBD-TT conjugated SARS-CoV-2 vaccine (PastoCovac). Immunity of the vaccinated volunteers was similarly evaluated at three-time points of baseline and after the first and second doses by a semiquantitative anti-spike serologic test. These control samples were taken at random from participants in the PastoCovac Phase 3 study at the Pasteur Institute of Iran.

### 2.6. Anti-SARS-CoV-2 Antibody Evaluation

The indirect-ELISA ChemoBind SARS-CoV-2 Neutralizing Antibody Test Kit (ChemoBind) is used to measure vaccine-induced antibody responses against the SARS-CoV-2 spike protein (anti-S) in both patients and healthy control groups. It evaluates total antibodies against the receptor-binding domain (RBD) of the Spike protein of SARS-CoV-2 using a semiquantitative Immunoassay. According to the manufacturer’s instructions, IgG Immune status ratio (ISR) values of <0.8 is negative, and >1.1 is positive; however, ratios between 0.8 and 1.1 are ambiguous and should be repeated.

### 2.7. Outcome

Serologic response at three weeks (±seven days) following the second dose of vaccine, defined as an increase in ISR to the cut-off value for the positive result (≥1.1) in the semiquantitative test, was considered the primary endpoint. Moreover, the strength of immunity was assessed by categorizing patients with moderate and strong immune responses after the second dose of the vaccine based on the median level of ISR raising [16].

As a secondary objective, the vaccine’s safety and tolerability were assessed up to 14 days after each dose was administered. Follow-up assessments are being carried out to evaluate the long-term safety and the stability of the immune response at least six months following immunization.

### 2.8. Safety Assessments

All reactogenicity events were recorded for seven days following each vaccination dosage by noting specific local (pain and redness at the injection site) or systemic (fever, fatigue, headache, diarrhea, vomiting, muscle pain) side effects described by patients. According to the Common Terminology Criteria for Adverse Events (CTCAE), all reactogenicity events were classified as none/mild (grade 0–1), moderate (grade 2), severe (grade 3), life-threatening/death (grade 4–5) [17]. Other non-reactogenicity adverse events were documented until two weeks following each vaccination dose delivery. Throughout the surveillance period, all vaccinated patients were kept under weekly telephone calls and at least every other week clinical visit to document any events such as a diagnosis of COVID-19, cytopenia, or recurrence of primary disease until 20 November 2022.

### 2.9. Statistical Analysis

All statistical analyses were performed using SPSS statistics software (version 23.0). Descriptive analysis was reported as mean with standard deviation (SD), median with interquartile range (IQR) for quantitative variables, and frequency with percent for qualitative variables. The normal distribution of ISRs was checked using the Shapiro–Wilk test. The Friedman test was used to test antibody titer differences during sampling and between groups. Antibody titer was compared between patients and controls for the three-time point sampling using the Mann–Whitney U test.

We used a logistic regression approach to investigate the predictive impact of selected baseline clinical parameters and laboratory indicators for strength of serologic response following the second vaccine dose based on the median level of ISR rising (Subtracting the baseline value from the post-second dosage yielded the ISR rising). In univariate analysis, predictors associated with strong immune response (*p* ≤ 0.20) are then incorporated into a multivariable logistic regression model using stepwise forward selection. All the tests were considered two-way, and a *p*-value < 0.05 was reported as statistically significant. The graphs were plotted by GraphPad Prism software (Version 8).

## 3. Results

### 3.1. Patient Characteristics

In this study, 38 auto-HSCT recipients who obtained two doses of PastoCovac and three blood samples were qualified to analyze the trial’s primary outcome at this stage of the trial (Figure 1). The study population included 16 (42.1%) females and 22 (57.9%) males with a mean (SD) age of 49.00 ± 11.54 at the time of auto-HSCT, respectively. The patients’ most important baseline characteristics and transplant details are presented in Table 1.

The main indication for auto-HSCT was multiple myeloma (MM) (52.6%) and lymphoma (47.4%). Peripheral blood was the graft source in all auto-HSCT recipients. The conditioning regimen was Melphalan for MM, and BEAM included Carmustine (BCNU), Etoposide, Ara-C, and Melphalan for lymphoma.

Our results showed the median (range) interval of 127 (90–271) days between auto-HSCT and the first dose and 32 (21–42) days between the first and the second doses. 21 (55.3%) of 38 HSCT recipients had a history of getting PCR-positive COVID-19 before transplantation. Most HSCT recipients (34 of 38) had received a conventional COVID-19 vaccination course before transplantation, as shown in Table 1.

### 3.2. Serological Outcomes

The mean ISR before the first dose of the vaccine was 1.39 (95% CI: 1.13–1.65) for auto-HSCT recipients and 1.91 (95% CI: 1.54–2.29) for healthy control people, indicating that 23 of 38 (60.5%) patients and 32 of 50 (64%) healthy volunteers had baseline ISR levels above the threshold for a positive result in the semiquantitative test. Figure 2 depicts a scatter plot of the SARS-CoV-2 IgG Immune status ratio (ISR) in patients and healthy controls over three time points. Figure 2 shows that the ISR was comparable between patients and healthy controls at each time point sampling.

After receiving the first and second doses, the mean ISR significantly increased compared to the baseline, reaching 2.47 (95%CI: 1.93–3.03) and 3.75 (95%CI: 3.13–4.38) (*p* < 0.001) in the auto-HSCT recipients and 2.75 (95%CI: 2.27–3.23) and 3.01 (95%CI: 2.56–3.46) (*p* < 0.001) in the healthy controls. As a result, the rate of seropositive tests rose to 81.6% and 93.3% in auto-HSCT recipients, as well as 90% and 96% in the healthy controls.

As depicted in Table 1, the mean (SD) ISR was compared between the three-time points of sampling. The values of ISR were higher in patients with a history of pre-HSCT PCR-positive COVID-19 (*p* = 0.046) and those who had received the pre-HSCT COVID-19 immunization (*p* = 0.040). However, no significant difference was demonstrated between age groups (≤40 vs. >40 years), sex groups, primary disease, the time interval between HSCT and vaccination (≤130 vs. >130 days), and baseline lymphocyte count (<1000 vs. ≥1000 cell/µ).

Compared to the baseline, two vaccine doses raised ISR by an average of 2.37 (SD: 1.92) in auto-HSCT recipients. As previously noted, we categorized patients with moderate and strong immune responses based on the value of ISR rising.

The logistic regression analysis of strong immune response predictors following the second dose of vaccine is presented in Table 2. In multivariate analysis, the higher count of lymphocytes [OR: 8.57 (95% CI: 1.51–48.75); *p* = 0.02] and history of getting COVID-19 before transplantation [OR: 6.24 (95% CI: 1.17–33.15); *p* = 0.03] remained the two independent positive predictors of the strong immune response following the second dose of vaccine.

Up to this stage of the ongoing trial and at a median follow-up of 165 (range 87–225) days from the second dose of vaccination till the last contact, five PCR-documented COVID-19 infections were reported among the patients who received the two doses of the post-HSCT vaccine, which presented with mild respiratory symptoms.

### 3.3. Safety

Data on vaccine-related adverse events are shown in Figure 3. No serious adverse events (≥Grade 3) were reported according to the CTCAE in any participant. The most frequent adverse event was pain at the injection site. Adverse events after the second dose were slightly more than those after the first dose in both types of HSCT. This study recorded no non-reactogenicity-related events until two weeks after each vaccine dose administration.

## 4. Discussion

We prospectively investigated the safety and serologic response of a new SARS-CoV-2 vaccine platform (RBD-TT conjugated SARS-CoV-2 vaccine) early after autologous transplants, compared to the control group, for the first time. Interestingly, 23 of 38 (60.5%) patients had positive anti-S1 ISR before vaccination despite a negative history of COVID-19. This finding concords with two recent studies in post-HSCT patients reported by Shah, G.L et al. and Majcherek M et al. [18,19] and could be explained by asymptomatic or mild COVID-19 before vaccination.

Following the first and second vaccine doses, 81.6% and 93.3% of all patients had positive ISR tests, respectively; our findings are consistent with those [20,21], who found a seroconversion rate of more than 90% after the second vaccine dosage in auto-HSCT recipients. It should be mentioned, however, that the median time between HSCT and immunization was longer in most previous trials than in ours. According to a meta-analysis, after two doses of COVID-19 vaccinations, the seropositive proportion for auto-HSCT patients was 81.9% (CI 95%; 64.3–91.9) [6].

Despite the short time between auto-HSCT and vaccination, the high seropositivity could be partially attributed to the structure and chemistry of our vaccine, as protein-conjugated antigens are more immunogenic than unconjugated antigens in HSCT patients [22]. According to Pao M et al. (2008), tetanus toxoid-conjugated vaccine platform and polysaccharide-protein conjugates (for example, Haemophilus influenzae type B vaccine) have a significant influence on immunological responses early after HSCT [23]. The conjugation of RBD to TT has been discovered to enhance cellular and humoral immune responses [14,15].

Because COVID-19 is prevalent in our area, most patients were vaccinated or naturally infected before HSCT. This may have led to the persistence of anti-SARS-CoV-2 antibodies in the early post-HSCT period and even better immune response to SARS-CoV-2 vaccination after HSCT. Our result supported this issue, exhibiting that pre-HSCT PCR positivity COVID-19 (*p* = 0.046) and pre-HSCT COVID-19 vaccination (*p* = 0.040) were associated with a higher serologic response to immunization early after transplantation.

The predictive value of pre-HSCT COVID-19 infection on stronger immune response was documented in the multivariate model. Consistently, the recent publication shows that prior infection significantly contributes to SARS-CoV-2-specific immunity and increases vaccination responses in a non-immunosuppressed cohort [24] and individuals over 80 [25].

We also found that pre-transplant COVID-19 vaccination would improve the immune response to post-HSCT SARS-CoV-2 vaccination. Along with our findings, Jullien M et al. showed the durability of anti-SARS-CoV-2 antibodies up to 9 months post-transplant in recipients who were immunized before HSCT [26]. In conclusion, contrary to existing guidelines [27], our findings support the concept of contemplating anti-SARS-CoV-2 vaccination before auto-HSCT.

In contrast to several previous studies [19,20,28,29], in which the time elapsed after HSCT was shown to be the most important factor determining the response, we found no correlation between the interval from transplant to immunization and seroconversion. The discrepancy may be explained by the fact that most recent studies enrolled patients over a wide range of time intervals between HSCT and vaccination. In contrast, our patients were vaccinated between 3 and 9 months following HSCT. Additionally, we discovered that a higher lymphocyte count (OR: 8.57, *p* = 0.02) appeared to be a reliable, independent positive predictor of a profound immunological response after the second dose of the vaccine [19,29,30].

Our study is constrained by the limited sample size and single-center design. We could not accurately estimate the antibody level using the semiquantitative test. Due to the lack of a validated assay to detect SARS-CoV-2-specific T-cell responses following vaccination, we were thus unable to evaluate cellular immunity.

## 5. Conclusions

Our findings indicate, for the first time to our knowledge, that the RBD-TT conjugated SARS-CoV-2 vaccine (PastoCovac) has shown significant safety and immunogenicity improvements after HSCT. Moreover, given that the RBD-TT conjugated SARS-CoV-2 vaccine could be manufactured and made more affordable on a large scale, these vaccines have become more relevant for underdeveloped countries. Our findings support the concept of contemplating anti-SARS-CoV-2 vaccination of recipients before HSCT; however, considering the limited number of patients in our research group, this conclusion should be regarded with care.

## Figures and Tables

**Figure 1 vaccines-11-00117-f001:**
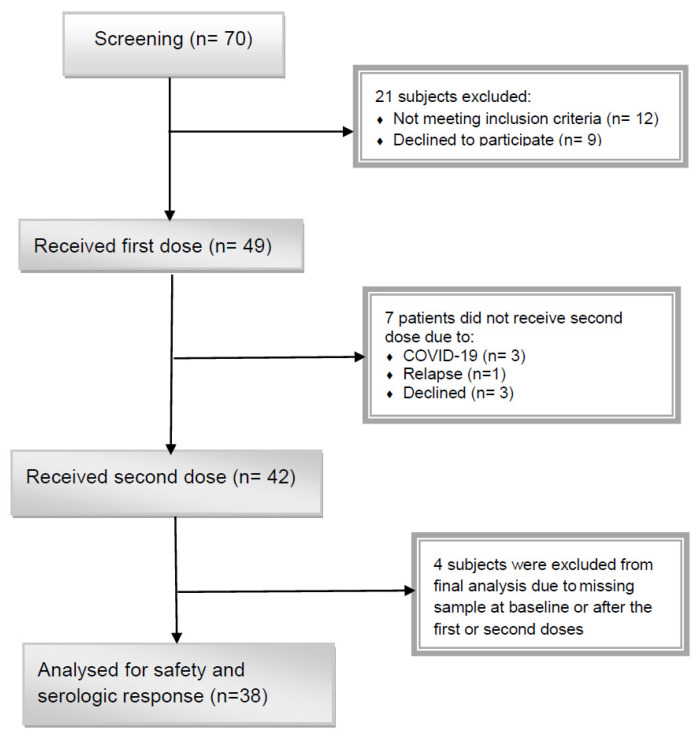
Flow chart of the qualification assessment process. A total of 70 participants were screened, and after adjusting for the inclusion and exclusion criteria, 38 subjects underwent analysis.

**Figure 2 vaccines-11-00117-f002:**
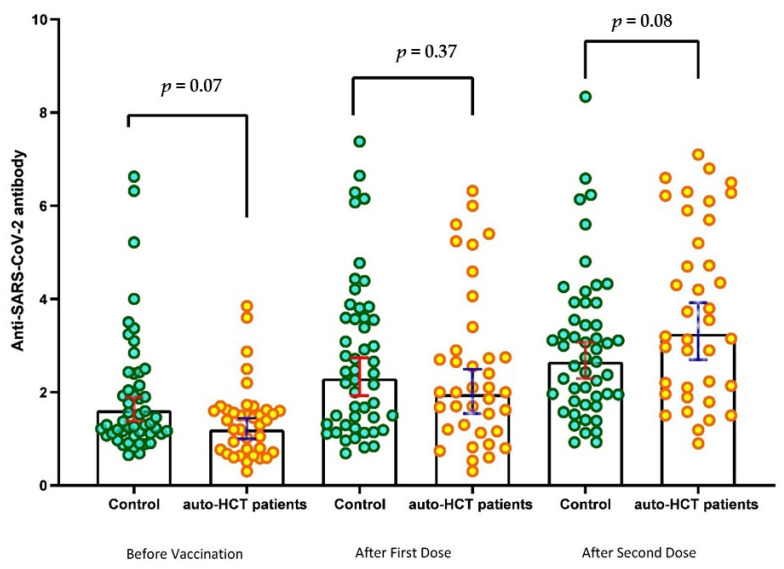
Scatter plot of SARS-CoV-2 IgG Immune status ratio (ISR) during the predefined sampling in 38 auto-HSCT recipients and 50 healthy controls.

**Figure 3 vaccines-11-00117-f003:**
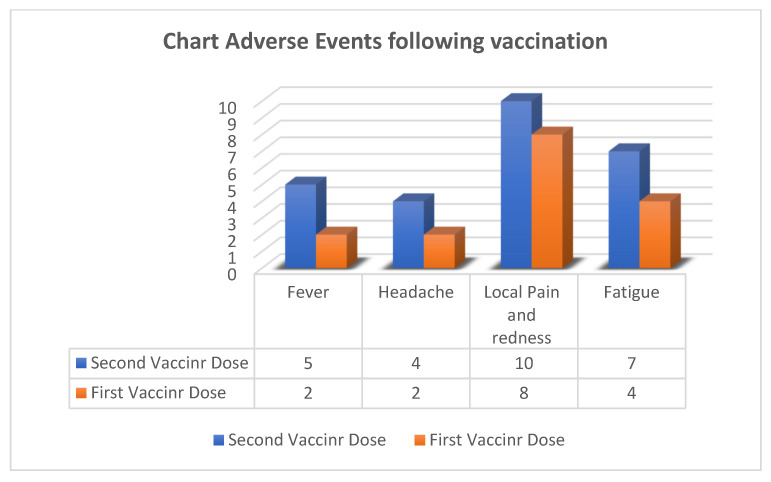
Number of adverse events following the first and second doses of vaccine in auto-HSCT recipients.

**Table 1 vaccines-11-00117-t001:** Serologic response after vaccination based on patients’ characteristics.

Arm	Characteristic		Number (%)	the Mean (SD) of ISR			
				Before First Dose	After First Dose	After Second Dose	*p* Value
Control Healthy Group	Sex	Female	22 (44)	1.92 ± 1.49	2.67 ± 1.77	3.02 ± 1.69	0.89
		Male	28 (56)	1.90 ± 1.21	2.81 ± 1.64	3.00 ± 1.51	
	Age (* Mean in years)	≤40	35 (70)	1.83 ± 1.26	2.83 ± 1.64	3.08 ± 1.43	0.68
		>40	15 (30)	2.09 ± 1.50	2.56 ± 1.82	2.85 ± 1.92	
	Total		50 (100)	1.91 ± 1.33	2.75 ± 1.68	3.01 ± 1.58	---
Auto-HCT Group	Sex	Female	16 (42.1)	1.40 ± 0.81	2.56 ± 1.65	3.94 ± 2.07	0.82
		Male	22 (57.9)	1.37 ± 0.80	2.42 ± 1.71	3.75 ± 1.89	
	Age (* Mean in years)	≤40	10 (24)	1.48 ± 0.97	2.39 ± 1.61	3.90 ± 2.10	0.78
		>40	28 (76)	1.29 ± 0.57	2.47 ± 1.67	3.61 ± 1.70	
	Background disease	Lymphoma	18 (47.4)	1.29 ± 0.81	2.49 ± 1.72	3.90 ± 2.11	0.87
		MM	20 (52.6)	1.47 ± 0.78	2.47 ± 1.67	3.61 ± 1.70	
	Lymphocyte Count (cells/µL)	<1000	16 (42.1)	1.26 ± 0.48	2.55 ± 1.90	3.14 ± 1.68	0.29
		≥1000	22 (57.9)	1.48 ± 0.96	2.43 ± 1.52	4.21 ± 1.95	
	Pre-HSCT COVID-19 vaccination	Yes	34 (89.5)	1.47 ± 0.80	2.59 ± 1.72	3.93 ± 1.90	0.04
		No	4 (10.5)	0.72 ± 0.15	1.49 ± 0.66	2.29 ± 1.12	
	Pre-HSCT PCR-positive COVID-19	Yes	21 (55.3)	1.50 ± 0.89	2.83 ± 1.68	4.21 ± 1.80	0.046
		No	17 (44.7)	1.25 ± 0.65	2.05 ± 1.60	3.20 ± 1.91	
	Median (range) time between HSCT and Vaccination in days	<130	19 (50)	1.44 ± 0.87	2.29 ± 1.49	3.58 ± 1.87	0.56
		≥130	19 (50)	1.34 ± 0.73	2.67 ± 1.67	3.93 ± 1.96	
	Total		38 (100)	1.39 ± 0.79	2.48 ± 1.67	3.75 ± 1.89	---
Total			88	1.68 ± 1.15	2.63 ± 1.67	3.33 ± 1.75	---

ISR, Immune Status Ratio; HSCT, hematopoietic stem cell transplant; MM, multiple myeloma. * The mean age of 40 was related to the total population (Patients and controls).

**Table 2 vaccines-11-00117-t002:** Univariate and Multivariate logistic Regression analysis of immunogenicity after two doses of the SARS-CoV-2 vaccine.

Effect	Univariate		Multivariate	
	Unadjusted OR (95% CI)	*p* Value	Adjusted OR (95% CI)	*p* Value
Patients Age (≥50.5 vs. <50.5)	0.80 (0.22–2.95)	0.74		
Patients Sex (Male vs. Female)	1.83 (0.47–7.07)	0.37		
Background disease (Lymphoma vs. MM)	0.67 (0.18–2.49)	0.55		
Pre-HSCT PCR-positive COVID-19 (Yes vs. No)	3.58 (0.87–14.65)	0.07	6.24 (1.17–33.15)	0.03
Pre-HSCT COVID-19 vaccination (Yes vs. No)		0.14		
Lymphocyte counts at vaccination (≥1000 vs. <1000)	5.20 (1.15–23.54)	0.03	8.57 (1.51–48.75)	0.02
Time between HSCT and Vaccination Median in Days (≥130 vs. <130)	1.95 (0.52–7.31)	0.32		

OR, odds ratio; CI, confidence interval; ISR, Immune status ratio; HSCT, hematopoietic stem cell transplant; MM, Multiple Myeloma.

## Data Availability

Data Availability Statements are available in section “MDPI Research Data Policies” at https://www.mdpi.com/ethics, accessed on 4 November 2022.

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
