# Peer review of "Evaluation of Safety and Immunogenicity of a Recombinant Receptor-Binding Domain (RBD)-Tetanus Toxoid (TT) Conjugated SARS-CoV-2 Vaccine (PastoCovac) in Recipients of Autologous Hematopoietic Stem Cell Transplantation Compared to the Healthy Controls; A Prospective, Open-Label Clinical Trial"

_vaccines, 2023, doi:10.3390/vaccines11010117_

Round 1

Reviewer 1 Report

The authors have evaluated the safety and immunogenicity of a recombinant 2 receptor-binding domain (RBD)-Tetanus Toxoid (TT) Conju- 3 gated SARS-CoV-2 vaccine (PastoCovac) in recipients of autologous hematopoietic stem cell transplantation.    

The finding is interesting, while some points need to be addressed well.

  1. Native speakers may further improve the language in the manuscript.
  2. Please report the key outcomes (OR value) with 95% CI in the Abstract. 
  3. Add more details for the critical and specific eligibility criteria in Materials and Methods.
  4. Please modify the format of Figure 1 to make it clear and organized.
  5. Please clarify if there is any active monitoring tool or mechanism to ensure data collection.
  6. Add OR value with 95% CI in the Result.
  7.  Please consider tabulating the number and percentage of adverse events resulting from safety assessment.

Author Response

Reply to the Review Report (Reviewer 1)

Thanks for your excellent comments

The request changes were performed with blue color in manuscript.

  1. Native speakers may further improve the language in the manuscript.
  2. Please report the key outcomes (OR value) with 95% CI in the Abstract. Was done in abstract
  3. Add more details for the critical and specific eligibility criteria in Materials and Methods. Were done in Inclusion criteria 

     4. Please modify the format of Figure 1 to make it clear and organized.

       It was corrected

      5. Please clarify if there is any active monitoring tool or mechanism to ensure data collection. Was added in the Safety assessment’s part, lines: 158-161

     6. Add OR value with 95% CI in the Result. Was done in results

      7. Please consider tabulating the number and percentage of adverse events resulting from safety assessment. The Figure 3 was changed to consider tabulating the number and percentage of adverse events

Reviewer 2 Report

As there is an urgent need for access to an effective and affordable vaccine for HSCT recipients in the endemic area, the authors investigated the safety and anti-spike serologic response of the new RBD-TT conjugated SARS-CoV-2 vaccine early after auto-HSCT. They found that the IgG ISR significantly increased after the first and second dosages of vaccine. The multivariate analysis showed that the higher count of lymphocytes and history of SARS-CoV-2 infection before transplantation could be positive predictors of the strong immune response following the second dose of vaccine. They also found that pre-transplant COVID-19 vaccination would improve the immune response to post-HSCT SARS-CoV-2 vaccination. Pain at the injection site was the most common adverse event, but any significant adverse effects were not observed in the participants. These data suggest that the RBD-TT conjugated SARS-CoV-2 vaccine is safe, highly immunogenic, and affordable early after autologous transplants. The findings are of considerable interest and the manuscript is well-written. However, I have raised several questions and minor points which need to be clarified. These are given below.

Specific comments:

1)     The authors emphasized the effectiveness and safety of the RBD-TT conjugated SARS-CoV-2 vaccine for HSCT recipients. However, this study contained no controls for comparison. I would like the authors to describe what this vaccine is superior to the mRNA vaccines and adenoviral vector vaccine for HSCT recipients, even if it is a comparison in the literature.

2)     Lines 202-204: The sentence “The values of ISR were higher in patients with a history of getting PCR-positive COVID-19 before transplantation (P= 0.046) and those who had received the pre-HSCT COVID-19 immunization (P= 0.040).” is not consistent with the data of “Vaccination history before HSTC” and “Covid-19 infection before HSCT (supposed to be PCR-positive COVID-19)” in Table 1. 

3)     According to the data in Table 1, COVID-19 infection before HSCT did not increase vaccination response. Is this data correct? And if so, what is the reason for this phenomenon? 

4)     Lines 285-288: “Our result supported this issue, exhibiting that prior COVID-19 PCR positivity before transplantation (P= 0.046) and pre-HSCT COVID-19 vaccination (P= 0.040) were associated with a higher serologic response to immunization early after transplantation.” Is there any data of “pre-HSCT COVID-19 vaccination (P= 0.040)” in Table 1?

5)     Minor points

(i)             Lines 74-77: The same sentence was written twice.

(ii)            Line 196: 2.48 should be 2.47?; line 197: 3.73 should be 3.75? (according to the data in Table 1)

(iii)          Lines 50-52: Is this one sentence?

Author Response

Reply to the Review Report (Reviewer 2)

Thanks for your excellent comments

  1. The authors emphasized the effectiveness and safety of the RBD-TT conjugated SARS-CoV-2 vaccine for HSCT recipients. However, this study contained no controls for comparison. I would like the authors to describe what this vaccine is superior to the mRNA vaccines and adenoviral vector vaccine for HSCT recipients, even if it is a comparison in the literature.

To fix this problem, we added a control group comprised of fifty healthy adults that were taken at random from participants in the PastoCovac Phase 3 study at the Pasteur Institute of Iran.

  1. Lines 202-204: The sentence “The values of ISR were higher in patients with a history of getting PCR-positive COVID-19 before transplantation (P= 0.046) and those who had received the pre-HSCT COVID-19 immunization (P= 0.040).” is not consistent with the data of “Vaccination history before HSTC” and “Covid-19 infection before HSCT (supposed to be PCR-positive COVID-19)” in Table 1. 

There was a mistake in Table 1 that has since been resolved.

  1. According to the data in Table 1, COVID-19 infection before HSCT did not increase vaccination response. Is this data correct? And if so, what is the reason for this phenomenon? 

There was a mistake in Table 1 that has since been resolved.

  1. Lines 285-288: “Our result supported this issue, exhibiting that prior COVID-19 PCR positivity before transplantation (P= 0.046) and pre-HSCT COVID-19 vaccination (P= 0.040) were associated with a higher serologic response to immunization early after transplantation.” Is there any data of “pre-HSCT COVID-19 vaccination (P= 0.040)” in Table 1?

There was a mistake in Table 1 that has since been resolved.

5)     Minor points

(i)             Lines 74-77: The same sentence was written twice.

                  This was made right.

(ii)            Line 196: 2.48 should be 2.47?; line 197: 3.73 should be 3.75? (according to the data in Table 1)           

                  This was made right.

(iii)          Lines 50-52: Is this one sentence?

                This has been fixed.

Reviewer 3 Report

The authors have conducted a an open label study to test the immunogenicity and tolerability of PastoCovac vaccine on recipients of hematopoetic stem cell transplantation. The authors  measured the IG levels after the first and second doses of the vaccine and showed that the second dose induced stronger responses than the first. Furthermore, the authors recorded only mild post vaccination adverse events and concluded that the vaccines tested was safe. I have a major concern: the study has no control group eg of healthy subjects who received the PastoCovac vaccine for comparison.The lack of a control group makes interpretation of the results difficult. For this reason I do not recommend the publication of this study  in its present form.

Author Response

To fix this problem, we added a control group comprised of fifty healthy adults that were taken at random from participants in the PastoCovac Phase 3 study at the Pasteur Institute of Iran.

Round 2

Reviewer 3 Report

I am satisfied with the revised version in which the authors have included a control group in the study design, making the results  convincing.

I therefore recommend the paper for for publication in MDPI Vaccines.